# Chromosomal Instability Is Associated with cGAS–STING Activation in EGFR-TKI Refractory Non-Small-Cell Lung Cancer

**DOI:** 10.3390/cells14060447

**Published:** 2025-03-17

**Authors:** Kimio Yonesaka, Takashi Kurosaki, Junko Tanizaki, Hisato Kawakami, Kaoru Tanaka, Osamu Maenishi, Shiki Takamura, Kazuko Sakai, Yasutaka Chiba, Takeshi Teramura, Hiroki Goto, Eri Otsuka, Hiroaki Okida, Masanori Funabashi, Yuuri Hashimoto, Kenji Hirotani, Yasuki Kamai, Takashi Kagari, Kazuto Nishio, Kazuhiro Kakimi, Hidetoshi Hayashi

**Affiliations:** 1Department of Medical Oncology, Kindai University Faculty of Medicine, Osaka 589-8511, Japan; 2Department of Medical Oncology, Kishiwada City Hospital, Osaka 589-8511, Japan; 3Department of Pathology, Kindai University Faculty of Medicine, Osaka 589-8511, Japan; shrsp@med.kindai.ac.jp; 4Department of Immunology, Kindai University Faculty of Medicine, Osaka 589-8511, Japankakimi@med.kindai.ac.jp (K.K.); 5Department of Genome Biology, Kindai University Faculty of Medicine, Osaka 589-8511, Japan; 6Clinical Research Center, Kindai University Hospital, Osaka 589-8511, Japan; chibay@med.kindai.ac.jp; 7Division of Cell Biology for Regenerative Medicine, Institute of Advanced Clinical Medicine, Kindai University Faculty of Medicine, Osaka 589-8511, Japan; 8Translational Research Department, Daiichi Sankyo RD Novare Co., Ltd., Tokyo 134-0081, Japanmasanori.funabashi@daiichisankyo.com (M.F.); 9Discovery Intelligence Research Laboratories, Research Function, R&D Division, Daiichi Sankyo Co., Ltd., Tokyo 103-0023, Japan; 10Early Clinical Development Department, Development Function, R&D Division, Daiichi Sankyo Co., Ltd., Tokyo 103-0023, Japan; 11Discovery Research Laboratories I, Research Function, R&D Division, Daiichi Sankyo Co., Ltd., Tokyo 103-0023, Japan; yasuki.kamai@daiichisankyo.com (Y.K.);

**Keywords:** EGFR-mutated non-small-cell lung cancer, epidermal growth factor receptor tyrosine kinase inhibitors, chromosomal instability

## Abstract

Epidermal growth factor receptor tyrosine kinase inhibitors (EGFR-TKIs) are standard therapies for *EGFR*-mutated non-small-cell lung cancer (NSCLC); however, their efficacy is inconsistent. Secondary mutations in the *EGFR* or other genes that lead to resistance have been identified, but resistance mechanisms have not been fully identified. Chromosomal instability (CIN) is a hallmark of cancer and results in genetic diversity. In this study, we demonstrated by transcriptomic analysis that CIN activates the cGAS–STING signaling pathway, which leads to EGFR-TKI refractoriness in a subset of *EGFR*-mutated NSCLC patients. Furthermore, *EGFR*-mutated H1975dnMCAK cells, which frequently underwent chromosomal mis-segregation, demonstrated refractoriness to the EGFR-TKI osimertinib compared to control cells. Second, H1975dnMCAK cells exhibited activation of cGAS–STING signaling and its downstream signaling, including tumor-promoting cytokine IL-6. Finally, chromosomally unstable *EGFR*-mutated NSCLC exhibited enhanced epithelial–mesenchymal transition (EMT). Blockade of cGAS–STING-TBK1 signaling reversed EMT, resulting in restored susceptibility to EGFR-TKIs in vitro and in vivo. These results suggest that CIN may lead to the activation of cGAS–STING signaling in some *EGFR*-mutated NSCLC, resulting in EMT-associated EGFR-TKI resistance.

## 1. Introduction

Epidermal growth factor receptor tyrosine kinase inhibitors (EGFR-TKIs) are the standard first-line therapy for patients with non-small-cell lung cancer (NSCLC) harboring EGFR-activating mutations [1,2,3]. EGFR-TKIs have achieved tumor regression in many *EGFR*-mutated NSCLC cases; however, all tumors eventually become resistant to EGFR-TKIs through various mechanisms, including secondary EGFR mutations and MET gene amplification [4,5,6]. Despite intensive research, resistance mechanisms have not been fully identified [7]. In addition, although most patients respond to EGFR-TKI treatment, the duration of the response is inconsistent between individual cases, and it is expected that pre-existing genetic abnormalities in cancer cells may affect the outcome of EGFR-TKI treatment [1,2,3]. For example, *TP53* mutations are genetic abnormalities consistently associated with poor outcomes of EGFR-TKIs in various studies, although the underlying mechanisms are not yet fully understood [8,9].

Inflammation is a hallmark of cancer, and a local immune response mediated by inflammatory cytokines within the tumor is considered to affect cancer progression and anti-cancer treatment effectiveness [10,11]. In particular, interleukin 6 expression has been associated with worse outcomes in patients with NSCLC treated with EGFR-TKIs, which may induce tumor refractoriness to EGFR-TKIs via JAK–STAT3 signaling activation or suppression of anti-tumor immune responses [12,13,14,15]. Additionally other inflammatory cytokines such as TGF-β and TNF have also been observed to influence EGFR-TKI efficacy [12,16,17]. However, it is not fully understood how the inflammatory response within tumors is actually regulated in *EGFR*-mutated NSCLC, and whether it influences the outcome of EGFR-TKI treatment.

Chromosomal instability (CIN) refers to ongoing errors in chromosome segregation during mitosis and is another hallmark of cancer [18]. CIN has been observed in various types of cancer cells and is thought to cause genetic instability in tumors [19,20,21]. CIN confers genomic plasticity to cancer cells, improving their survival through the acquisition of karyotypes, and is therefore a target for the development of anti-cancer drug therapy [22,23]. CIN is also involved in the regulation of the innate immunity. Specifically, chromosome mis-segregation forms micronuclei, which accumulate cytosolic DNA upon its decay outside the nucleus [18,21]. Aberrant cytosolic DNA is recognized by cytoplasmic DNA receptors such as cyclic GMP-AMP synthase (cGAS), which in turn activates stimulator of interferon genes (STING) via the second messenger cGAMP [18,21]. STING activation complexed with TBK1 leads to the transcriptional activation of type I interferon (IFN) and inflammatory cytokines mediated by NF-κB [18,21,24]. Originally, the cGAS–STING pathway was identified as a mechanism that protects the host from viral infection by activating IFN signaling [25], but in cancer, activation of this process by CIN may lead to tumor cell death [24]. Moreover, several STING agonists are in clinical development for cancer therapy, exploiting the innate anti-tumor immune responses occurring via STING stimulation, which potentially enhances the efficacy of immune checkpoint inhibitor treatment [26,27]. Conversely, persistent activation of STING pathway by CIN can cause chronic inflammation, which in turn promotes cancer metastasis [21,28]. Thus, the activation of STING by CIN may have conflicting clinical consequences, depending on the circumstances.

In this study, we performed genomic and transcriptomic analyses of *EGFR*-mutated NSCLC tumor and cell lines to evaluate the relationship between CIN and the effect of EGFR-TKIs as well as the activity of the intrinsic cGAS–STING signaling pathway.

## 2. Materials and Methods

### 2.1. Study Design and Acquisition of Tumor Samples and Clinical Data

This retrospective study was based on genomic and transcriptomic analyses of 67 tissue samples obtained from patients with EGFR-mutated NSCLC before EGFR-TKI treatment (*n* = 32) and after the acquisition of EGFR-TKI resistance (*n* = 35) [29]. Only one re-biopsy was performed in each case after the acquisition of resistance. Twenty-three cases were paired specimens before and after EGFR-TKI treatment. The current data set was collected from a previous study [29]. All patients received EGFR-TKIs as the 1st-line therapy for advanced stage. The characteristics of patients are shown in Appendix A. Samples were preserved in formalin-fixed, paraffin-embedded (FFPE) tissue sections of tumors obtained from patients with EGFR-mutated NSCLC treated at Kindai Hospital, Kishiwada Hospital, and Izumi Hospital (Osaka, Japan) between 2018 and 2020. Most tumor samples are obtained from the lungs via bronchoscopic diagnostic biopsy or surgical resection. The re-biopsy site and clinical course are shown in Appendix A. Clinical data were collected from the medical records of all patients. This study was approved by the Institutional Review Boards of Kindai Hospital, Kishiwada Hospital, and Izumi Hospital. Written informed consent was obtained from all participants after the opportunity to opt out was provided.

### 2.2. DNA and RNA Extraction

As previously described [29], DNA and RNA were extracted [29]. In brief, FFPE specimens were subjected to histological examination, and only those specimens containing a sufficient quantity of tumor cells (≥50% for RNA, ≥30% for DNA) were subjected to nucleic acid extraction. The minimum absolute number of tumor cells that is required for the subsequent analysis is 200. DNA and RNA were purified using either the FormaPure system (Beckman Coulter Inc., Brea, CA, USA) or Allprep DNA/RNA FFPE kit (Qiagen, Valencia, CA, USA). A DNA input of 110 ng was utilized for library preparation. An RNA input of 20 ng was utilized for library preparation. DNA panel sequencing, RNA panel sequencing, and transcriptional profiling are described in the Appendix A.

### 2.3. Molecular Pathway Analysis

Gene set enrichment analysis (RRID:SCR_003199) was conducted using GSEA version 4.2.2 [30] with default settings. The gene set database selected was Hallmarks h.all.v2023.2.symbol.gmt. The gene set for assessing STING activation was generated in accordance with a previous report [31]. The data were sorted according to the normalized enrichment score and selected categories.

### 2.4. Computational Methods for Analysis of RNA-Seq Data and CIN Score

RNA-Seq data were analyzed using the following web tool: GenePattern 3.9.11 (RRID:SCR_003201) for ssGSEA. Subsequently, ssGSEA was performed using the gene sets of h.all.v2023.2.symbol.gmt as a gene set database. The CIN score was calculated in accordance with a previously reported methodology, using the mean of the normalized gene expression data for the 70 genes that comprise the signature [19].

### 2.5. Vector Engineering and Transfection

For fluorescence imaging of chromosomes, a Piggybac-transposon plasmid (pPB) containing cDNA of human histone H2B (hH2B)-mCherry fusion proteins was designed. To induce CIN, cDNA encoding alfa-tagged dnMCAK was inserted into the pPB-CAG vector followed by the brasticidin resistance gene with an internal ribosomal entry site sequence. Plasmids were constructed by VectorBuilder Inc. (Japan Headquarters, Kanagawa, Japan).

H1975 cells harboring a EGFR L858R/T790M double mutation (RRID:CVCL_1511) were co-transfected with the pPB plasmids and the helper plasmid encoding piggybac transposase (PBase) by electroporation using a NEPA21 electroporator, and then cultured in 10% FCS-RPMI 1640 medium (Thermo Fisher, Waltham, MA, USA) with 5 µg/mL blasticidin (Thermo Fisher) and 1 µg/mL puromycin (Thermo Fisher) for 7 days.

### 2.6. Imaging of Spindle Morphology and CIN

Prior to fluorescence microscopy, hH2B-mCherry/dnMCAK-expressing H1975 cells were seeded in glass-bottom dishes (Matsunami Glass Ind., Osaka, Japan) and treated with nocodazole (Sigma-Aldrich, St. Louis, MO, USA) at a final concentration of 10 µg/mL for 12 h, followed by 4 h of recovery culture. Spindle morphology was observed using a confocal laser microscope equipped with an oil-immersion objective lens (FV-3000; Olympus, Tokyo, Japan).

### 2.7. Immunoblotting

Cells were seeded at a density of 3 × 10^6^ cells per 90 mm plate, incubated overnight in RPMI-1640 medium containing 2% FBS, and collected. Western blotting was performed as previously described [29]. Proteins were examined using antibodies specific for phospho-EGFR (Cell Signaling Technology, Danvers, MA, USA, Cat# 2234, RRID:AB_331701), E-cadherin (Cell Signaling Technology Cat# 3195, RRID:AB_2291471), vimentin (Cell Signaling Technology Cat# 3390, RRID:AB_2216128), actin (Sigma-Aldrich Cat# A5441, RRID:AB_476744), and EGFR (Santa Cruz Biotechnology Cat# sc-03, RRID:AB_631420).

### 2.8. Colony-Formation Assay

H1975Cont and H1975dnMCAK cells were seeded in 6-well plates at a density of 3.0 × 10^4^ cells/well. After 24 h, cells were treated with 10 µM G150, a selective cGAS inhibitor, 0.5 µM C176, a selective STING inhibitor, 1 µM MRT67307, an IKKε and TBK-1 inhibitor, 0.1 µM osimertinib, or a combination of these drugs. The medium containing each drug was changed every 3 days. After 10 days of incubation, the plates were gently washed with phosphate-buffered saline (PBS) and fixed with a fixation solution (acetic acid/methanol 1:7) for 5 min. Colonies were stained with 0.5% crystal violet for 3 h at room temperature. The percentage of colony area was automatically calculated using Image J 1.52a (National Institutes of Health, Bethesda, MD, USA, RRID:SCR_003070).

### 2.9. In Vivo Tumor Growth Inhibition Assay

All animal experiments were performed in accordance with the Recommendations for Handling of Laboratory Animals for Biomedical Research compiled by the Committee on Safety and Ethical Handling Regulations or Laboratory Animal Experiments of Kindai University. The study protocol was reviewed and approved by the Animal Ethics Committee of Kindai University. H1975Cont or H1975dnMCAK cells (5 × 10^6^ cells/mouse) were subcutaneously injected into the right flanks of female BALB/cAJcl-nu/nu mice (CLEA Japan, Tokyo, Japan). Once tumors had reached the target volume (0.2 cm^3^), mice were randomly assigned to the treatment and control groups. Mice received vehicle (200 μL; control) or G150 (10 mg/kg body weight in 200 μL vehicle) by i.p. injection every two days, p.o. administration of osimertinib (5 mg/kg body weight in 100 μL PBS) five times per week, or both. The tumor volume and mouse body weight were measured twice per week. The mice were sacrificed if the tumors became necrotic or grew to a volume of 2.0 cm^3^. The tumor volume was defined as 1/2 × length × width^2^. The T/C ratio was calculated using the following equation: 100 × (average tumor volume of the treated group)/(average tumor volume of the control group).

### 2.10. Immunohistochemical Staining

Vimentin expression was evaluated by immunohistochemistry in EGFR-mutated NSCLC tissues. Sections (thickness, 4 μm) were immunologically stained for vimentin uisng an anti-vimentin mouse monoclonal antibody (Leica Biosystems Newcastle Ltd., Newcastle upon Tyne, UK) and an automated slide stainer, DAKO Link48 (Agilent Technologies, Inc., Santa Clara, CA, USA). The immunohistochemical expression of vimentin in the cancer cells was interpreted by a pathologist (O. M.).

### 2.11. Statistical Analysis

Statistical analyses were performed using the SPSS software (version 22.0; IBM Corp., Armonk, NY, USA, RRID:SCR_002865). Progression-free survival (PFS) was defined as the duration from the initiation of EGFR-TKI therapy until tumor progression or death from any cause. Kaplan–Meier curves were generated for PFS and used to calculate the median and 95% confidence interval (CI) of each treatment group. Two-sided *p* values were determined using the log-rank test, and hazard ratios (HRs) (95% CIs) were estimated using the Cox proportional hazards model. Pearson’s correlation coefficients (r) were calculated to explore the relationship between the CIN score and other genomic characteristics of the tumors. Differences in genomic expression were analyzed based on these differences. Other statistical tests were two-sided, unpaired *t*-tests, and *p*-values < 0.05 were considered statistically significant. Graphical depictions of the data were obtained using GraphPad Prism 5.0 for Windows (GraphPad Software, Inc., Boston, MA, USA, RRID:SCR_002798).

## 3. Results

### 3.1. Influence of CIN on Outcomes of EGFR-Mutated NSCLC Treated with EGFR-TKIs

A total of 67 samples were obtained before EGFR-TKI treatment and after the acquisition of EGFR-TKI resistance, designated as the “pre-treatment” samples (*n* = 32) and “post-treatment” samples (*n* = 35), respectively, from patients with *EGFR*-mutated NSCLC whose tumors had initially responded to EGFR-TKI (Figure 1A). Posttreatment samples acquired secondary genetic alterations, including *EGFR* T790M, *HER2* amplification, *MET* amplification, or small cell carcinoma transformation, but almost half of the samples did not show genetic abnormalities causing EGFR-TKI resistance (Appendix A). We examined the relationship between CIN and several clinical characteristics, including sex, smoking status, *EGFR* mutation, *EGFR* secondary T790M mutation, and *TP53* mutation (Figure 1B). Of these backgrounds, *TP53* mutation in pretreatment samples was significantly correlated with CIN score (multiple *t*-tests; *p* = 0.01, Figure 1B). The relationship between CIN and EGFR-TKI treatment outcomes was evaluated by separating the median CIN scores into two groups, CIN-high and CIN-low, and by comparing progression-free survival (PFS) between the two groups. There was no obvious difference in patient background (Appendix A); however, PFS with EGFR-TKI treatment was worse in the CIN-high group than in the CIN-low group (HR 3.274; *p* = 0.005; median PFS 16.7 vs. 11.0 months, respectively; Figure 1C). When the association between patient backgrounds and PFS with EGFR-TKI treatment was evaluated, CIN and *TP53* mutations were found to be associated with PFS. Multivariate analysis showed that, for treatment with EGFR-TKIs, only CIN was associated with PFS (*p* = 0.038). The CIN score was also calculated in another cohort of *EGFR*-mutated NSCLC patients to evaluate the impact of CIN on PFS with EGFR-TKI treatment (Appendix A). The PFS in the CIN-high group was shorter than that in the CIN-low group (HR 4.838; *p* = 0.04, Appendix A). We then compared CIN in tumors before EGFR-TKI treatment and after the acquisition of EGFR-TKI resistance. The enrichment score for CIN was more pronounced after the acquisition of EGFR-TKI resistance than before EGFR-TKI treatment (normalized enrichment score, −1.60; *p* = 0.036; Figure 1E). The individual tumor comparison also showed that the CIN z-score was numerically higher in tumors that acquired EGFR-TKI resistance compared to tumors before EGFR-TKI treatment (−0.223 vs. 0.204, respectively, Figure 1F).

These results suggest that CIN is adversely associated with the outcome of EGFR-TKI treatment in patients with *EGFR*-mutated NSCLC, which could be independent of other factors influencing the treatment response, including *TP53* and *EGFR* mutations. Alternatively, chromosomally unstable cancer cells may become refractory to EGFR-TKIs.

### 3.2. CIN Activates an Innate Immune Response in EGFR-Mutated NSCLC

Next, we evaluated the association between CIN and tumor hallmark gene signatures after the development of EGFR-TKI resistance by gene set enrichment analysis (Figure 2A and Appendix A) [29]. Tumors with high CIN scores showed pronounced signatures of cancer cell proliferation and activated PI3K/AKT/mTOR signaling compared with tumors with low CIN scores (Appendix A). Furthermore, the activation of multiple inflammation-related signatures, including the inflammatory response, IFN responses, and IL6/JAK/STAT3 signaling, was observed in tumors with high CIN scores, especially in subpopulations with unknown causes of resistance to EGFR-TKIs, compared to tumors with low CIN scores (Figure 2A). A similar relationship between CIN and inflammation-related signatures was also observed in tumors before EGFR-TKI treatment but was further pronounced after the acquisition of resistance (Appendix A).

We then performed a transcriptome analysis examining the signaling pathway in *EGFR*-mutated NSCLC tumors developing EGFR-TKI resistance and explored the underlying mechanism of the CIN-related inflammatory response. The CIN score correlated with the expression level of cytoplasmic DNA sensors, including cGAS, and with STING activation in tumors, particularly with unknown causes of EGFR-TKI resistance (Figure 2B). Furthermore, CIN correlated with the activity of TANK-binding kinase 1 (TBK1), a binding partner of STING, and its substrate, interferon regulatory factor 3 (IRF3), as well as with type I IFN signaling (Figure 2B). Finally, CIN correlated with the activity of NF-κB and IL-6 signaling in these tumors (Figure 2B).

Collectively, these results suggest that CIN may activate an innate immune response via the cGAS–STING and downstream inflammatory cytokine and NF-κB signaling pathways in *EGFR*-mutated NSCLC tumors (Figure 2C). Therefore, chromosomally unstable tumors have anti-tumor properties, including activation of cytotoxic type I IFN signaling and anti-tumor immunity; however, they also have tumor-promoting properties, associated with NF-κB and IL-6 signaling (Figure 2C). It is unclear how these conflicting properties affect the susceptibility of *EGFR*-mutated NSCLC cells to EGFR-TKIs.

### 3.3. Chromosomal Instability Induces EGFR-TKI Refractoriness

We tested whether CIN interfered with the susceptibility of *EGFR*-mutated NSCLC cells to EGFR-TKIs. Previously, the induction of dominantly negative *MCAK* has been reported to increase chromosomal instability in cancer cells [32]. When dominant negative *MCAK* was induced in the *EGFR*-mutated NSCLC H1975 cell line, mitotic abnormalities, including chromosomal lagging, bridging, and isolation, were observed more frequently during cell division in H1975dnMCAK cells than in control H1975Cont cells (Figure 3A,B). In vitro sensitivity assays demonstrated that exposure to the EGFR-TKI osimertinib for 72 h decreased the viable cell count in a concentration-dependent manner in H1975Cont cells but maintained a viable cell count in H1975dnMCAK cells (Figure 3C). In a mouse xenograft model study, H1975Cont tumors shrunk upon osimertinib exposure, whereas H1975dnMCAK tumors did not shrink and continued to grow for three weeks (Figure 3D).

The effects of osimertinib on intracellular signaling were evaluated in chromosomally unstable NSCLC cells. In both H1975Cont and H1975dnMCAK cells, osimertinib decreased the autophosphorylation of EGFR and its downstream ERK phosphorylation (Figure 3E). In contrast, STAT3 phosphorylation was greater in H1975dnMCAK cells than in H1975Cont cells and was maintained under EGFR-TKI exposure (Figure 3E). Because activated STAT3 potentially induces epithelial–mesenchymal transition (EMT), we evaluated the expression of EMT transcription factors (EMT-TFs) and epithelial or mesenchymal markers in H1975Cont and H1975dnMCAK clone cells. The H1975dnMCAK clones showed a decrease in the expression of the epithelial marker CDH1 compared to the H1975Cont clones, whereas the H1975dnMCAK clones showed an increase in EMT or EMT-TF markers including ZEB1,2, SNAI1,2, and VIM and AXL (Figure 3F) [33]. In addition, H1975dnMCAK cells showed decreased expression of the epithelial marker E-cadherin and increased EMT-TFs snai1, zeb1 protein, and the mesenchymal marker vimentin protein compared to H1975Cont cells (Figure 3G).

These results suggest that CIN may be associated with EMT in *EGFR*-mutated NSCLC, which interferes with the efficacy of EGFR-TKIs. However, the mechanisms underlying CIN-associated EMT remains unclear.

### 3.4. EMT Is Associated with Activated cGAS–STING Signaling in Chromosomally Unstable NSCLC with EGFR-Activating Mutation

Next, we tested whether EMT was associated with the activation of cGAS–STING signaling in H1975dnMCAK cells. Transcriptomic analysis revealed that the STING signal and its downstream type I IFN and IL-6 signals were activated in the H1975dnMCAK clones compared to the H1975Cont clones (Figure 4A). Phosphorylation of TBK1 also increased in H1975dnMCAK cells compared to that in the H1975Cont cells (Figure 4B).

Then, H1975Cont and H1975dnCAK cells were exposed to the cGAS inhibitor G150 to evaluate its effect on EMT [34]. The expression of the epithelial marker E-cadherin was lower in H1975dnMCAK cells than in H1975Cont cells (Figure 4C). H1975dnMCAK cells maintained low levels of E-cadherin expression, even after exposure to either G150 alone or in combination with osimertinib (Figure 4C). In contrast to E-cadherin, the expression of vimentin, a mesenchymal marker, was greater in H1975dnMCAK cells than in H1975Cont cells, but decreased after exposure to either G150 alone or in combination with osimertinib in H1975dnMCAK cells (Figure 4C). To evaluate the relationship between CIN-associated EMT and osimertinib susceptibility in H1975dnMCAK cells, a colony-formation assay was performed. In H1975Cont cells, osimertinib exposure decreased colony formation to 24% of that of the DMSO control (Figure 4D). In contrast, the effect of osimertinib in H1975dnMCAK cells was more limited, decreasing colony formation to 44% of the DMSO control level, and G150 exposure decreasing colony formation to 64% of that of the DMSO control (Figure 4E). However, osimertinib in combination with G150 reduced colony formation to 10% that of the DMSO control (Figure 4E). Additionally, we evaluated the influence of TBK1, which is downstream of STING signaling, on EMT in H1975dnMCAK cells using the TBK1 inhibitor MRT67307. Similar to cGAS inhibition, TBK1 inhibition had a limited effect on the expression of epithelial marker E-cadherin, either alone or in combination with osimertinib (Figure 4F). However, TBK1 inhibition decreased the expression of vimentin in H1975dnMCAK cells, either alone or in combination with osimertinib, as observed with cGAS inhibition (Figure 4F). Colony-formation assays using H1975dnMCAK cells showed that osimertinib and MRT67307 decreased colony formation to 44% and 38% of the DMSO control level, respectively. However, the combination of osimertinib and MRT67307 decreased colony formation to 18% of the DMSO control level (Figure 4G).

Next, to specifically inhibit cGAS–STING signaling, we knocked down cGAS using siRNA and examined its effect on EMT in H1975dnMCAK cells. cGAS knockdown decreased the phosphorylation of TBK1 compared to control siRNA and reduced the expression of EMT-TFs snal1, zeb1, and the mesenchymal marker vimentin in H1975dnMCAK cells (Figure 4H). In addition, the effect of cGAS knockdown in H1975dnMCAK cells on their susceptibility to 100 nM osimertinib for 3 days was evaluated. Neither osimertinib exposure alone nor cGAS knockdown alone had a significant effect on cell viability (Figure 4I). However, osimertinib exposure and cGAS knockdown resulted in a significant decrease in cell viability (Dunnett’s multiple comparison test, *p* < 0.0001; Figure 4I).

Additionally, we examined the efficacy of the cGAS inhibitor, G150, and its combination with osimertinib in a xenograft mouse model using chromosomally unstable H1975dnMCAK cells. The anti-tumor efficacy of G150 or osimertinib alone was limited, whereas the drug combination significantly inhibited tumor growth compared to the control (multiple comparison test; *p* = 0.021, Figure 4J).

These results suggest that CIN is associated with EMT via activation of the cGAS–STING–TBK1 signaling pathway, potentially resulting in EGFR-TKI refractoriness in *EGFR*-mutated NSCLC. Blockade of the cGAS–STING–TBK1 pathway may reverse EMT and restore susceptibility to EGFR-TKIs in chromosomally unstable NSCLC with activating *EGFR* mutations.

### 3.5. Epithelial–Mesenchymal Plasticity Observed in Chromosomally Unstable NSCLC Tumor Harboring EGFR-Activating Mutation

Next, we assessed the relationship between EMT and CIN in *EGFR*-mutated NSCLC tumors that were refractory to EGFR-TKIs with unknown causes of resistance. Gene set enrichment analysis showed a moderate correlation between the CIN score and hallmark EMT gene signature in these tumors (Figure 5A). According to individual gene expression, epithelial markers, including CDH1, did not correlate with CIN in these refractory tumors (Figure 5B). However, the expression levels of EMT transcription factors, including SNAI1 and mesenchymal markers, including vimentin and AXL, correlated with CIN scores (Figure 5B) [33,35,36,37]. Furthermore, vimentin expression was assessed by immunohistochemical evaluation of a subpopulation of chromosomally unstable NSCLC tumors (Figure 5C,D). The highest expression of vimentin was observed in tumor #9, which had the highest CIN score among these tumors, and partial vimentin expression was observed in tumors with moderate CIN, including tumors #20 and #34 (Figure 5C,D). CIN was correlated with the ratio of vimentin-positive cells in tumors (Pearson, r = 0.648).

Collectively, these results suggest that epithelial–mesenchymal plasticity is associated with cGAS–STING activation in chromosomally unstable NSCLC with EGFR-activating mutations and may limit the clinical outcome of EGFR-TKI treatment (Figure 5E).

## 4. Discussion

In this study, we combined analyses of clinical tumor samples and the use of a cell line to evaluate the relationship between CIN and the effect of EGFR-TKIs in *EGFR*-mutated NSCLC. CIN is associated with restricted clinical outcomes of EGFR-TKI treatment using the activated cGAS–STING pathway. Previously, Jin, et al. reported that the chromosome instability score (CIS), defined as the proportion of the genome with an aberrant segmented copy number, increased in *EGFR*-mutated NSCLC tumors after developing resistance to EGFR-TKIs compared to before treatment [38]. In particular, the increase in CIS in *EGFR* T790M-negative tumors was more pronounced than that in EGFR T790M-positive tumors. In addition, acquired somatic copy number abnormalities in several genes involved in the maintenance of CIN, such as *TP53* and *AURKA*, appear to be more advanced in *EGFR* T790M-negative tumors than in T790M-positive tumors [38]. The present study also observed a notable increase in CIN in tumors that had developed resistance to EGFR-TKIs compared to that before treatment (Figure 1E,F). Moreover, CIN correlated with enhanced STING activation in tumors, particularly in the absence of identified resistance mechanisms to EGFR-TKIs (Figure 2B). Previous studies on resistance to EGFR-TKIs have frequently identified single genomic alterations, such as the secondary *EGFR* T790M mutation or *MET* amplification [5,7]. Nevertheless, in more than half of the resistant cases, no specific genetic abnormality was identified, suggesting that alternative mechanisms may be involved in maintaining cell proliferation independent of specific genetic abnormalities [39]. Given that CIN and associated cGAS/STING activation are enhanced in tumors lacking defined resistance mechanisms, it may be hypothesized that CIN is associated with EGFR-TKI resistance, independent of specific genomic alterations.

Previous studies have indicated that CIN contributes to karyotypic diversity in cancer cells, thereby facilitating resistance to various anti-cancer drugs [40]. Moreover, the results of the present study suggest that CIN-dependent activation of cGAS–STING signaling may contribute to cancer cell proliferation under EGFR-TKI treatment in a subset of *EGFR*-mutated NSCLC tumors. Nevertheless, the activation of cGAS–STING signaling may also impede tumor growth by activating type 1 IFN and concomitantly promoting anti-tumor immunity (Figure 2). Indeed, a number of STING stimulators are currently in clinical development for anti-cancer therapy [26,27]. Other investigators have reported that STING stimulation with STING agonists or irradiation improves the tumor microenvironment and sensitivity to EGFR-TKIs and immune checkpoint inhibitors in *EGFR*-mutated mouse tumor models [41,42,43]. One potential explanation for this discrepancy is the possibility of intrinsic suppression of type I interferon signaling in CIN-dependent persistent STING-activated tumors. Indeed, Bakhoum, et al. identified that cGAS–STING activation fails to promote type I interferon signaling or interferon-stimulated gene induction in MDA-MB-231 cells [18,21]. To evade cell-intrinsic type I interferon production, the interferon gene cluster on chromosome 9p may be lost in these tumors [44]. In fact, in the present study, the deletion of chromosome 9p was also observed in a subset of high-CIN tumors, including cases #38 and #13. Moreover, research conducted by Li et al. demonstrated that the persistent activation of the cGAS–STING pathway, which is induced by CIN, results in downstream signal rewiring in cancer cells [28]. This phenomenon is characterized by type I interferon tachyphylaxis downstream of the STING [28].

The present study demonstrated a correlation between CIN and EMT in H1975dnMCAK cells and in a subset of *EGFR*-mutated NSCLC tumors. The occurrence of EGFR-TKI resistance associated with EMT has been reported on numerous occasions, and therapeutic studies have sought to identify vulnerabilities that can be exploited to overcome resistance [12,13,36,45]. In particular, the EMT transcriptional regulator ZEB1 has been demonstrated to reduce E-cadherin and increase vimentin in multiple *EGFR*-mutated NSCLC cells, thereby conferring resistance to EGFR-TKIs [36]. SNAIL was reported to facilitate drug resistance and EMT [46]. In the present study, ZEB1 or SNAIL expression was also upregulated in H1975dnMCAK cells, which may be a contributing factor to EMT and may also be associated with EGFR-TKI resistance (Figure 3G). Additionally, activation of IL-6 and JAK/STAT3 signaling pathways has also been reported to be associated with EMT [12,15,47]. In the present study, we observed that these signaling pathways were activated in H1975dnMCAK cells and tumor tissues (Figure 2A,B and Figure 4A). Accordingly, we investigated whether the IL-6 inhibitor tocilizumab or the JAK inhibitor ruxolitinib could suppress EMT and overcome EGFR-TKI resistance, although these strategies proved to be ineffective (Appendix A). These results indicate that CIN-associated EMT may be regulated by multiple signaling pathways in *EGFR*-mutated NSCLC. In particular, STING has been demonstrated to stimulate NF-κB, which is known to affect the expression of several genes associated with EMT [48,49,50]. Indeed, the expression of multiple genes associated with EMT and NF-κB signaling was greater in H1975dnMCAK cells than H1975Cont cells (Appendix A). For example, Huber et al. reported that TGF-β-dependent induction of EMT depends at least in part on NF-κB activity [49]. The *TGFB1* was expressed more in H1975dnMCAK cells than in H1975Cont cells and may be involved in EMT (Appendix A).

In the current study, CIN or its downstream cGAS or STING expression was correlated with the clinical outcome of EGFR-TKI treatment (Figure 1C and Appendix A). cGAS mRNA expression was correlated with CIN in *EGFR*-mutated non-small-cell lung cancer (Figure 2B). Reportedly, cGAS is an IFN-stimulated gene, and its overexpression is associated with Type I IFN activation [51]. This suggests that cGAS expression may serve as a surrogate marker for CIN that potentially activates Type I IFN. Additionally, STING mRNA expression showed an inverse correlation with CIN, suggesting a potential negative regulation of STING mRNA expression by excessive CIN-induced STING activation (Appendix A). Thus, although hypothetical, STING mRNA expression may be a surrogated marker for CIN. These findings have implications for the development of biomarkers and novel therapeutic strategies for *EGFR*-mutated NSCLC. However, as the present study was retrospective in nature, further clinical validation is required to verify the findings in *EGFR*-mutated NSCLC. The development of a method for the accurate assessment of CIN is also necessary. Although the expression-based scoring method used in the present study may prove beneficial, the influence of mixed normal cells cannot be excluded. The evaluation of gene copy number using single-cell analysis may be feasible at the research level; however, it may prove challenging to apply in practical medicine due to the limited cellular resolution of tissue specimens or a cost issue [52]. Alternatively, the use of micronuclei in clinical practice has been established for the assessment of radiation-induced effects, and there is potential for their use in the assessment of CIN [53]. Additionally, CIN-dependent cGAS–STING signaling activity may influence the tumor immune environment, particularly with regard to inflammatory cell infiltration, which may consequently influence the efficacy of EGFR-TKI therapy [11,28,41]. Yet, the current study did not assess the influence of CIN-dependent cGAS–STING signaling activity in *EGFR*-mutated NSCLC on the tumor immune environment or the efficacy of EGFR-TKI therapy. This topic warrants further investigation in future studies. Another limitation is that *dnMCAK* induction was successful in transducing CIN into the H1975 cell line, but not into other cell lines. For further studies, alternative methods of transducing CIN into multiple cell lines need to be tested. Despite some limitations, STING or cGAS inhibitors are currently in clinical development for inflammatory diseases and may be applicable to *EGFR*-mutated NSCLC [54].

## Figures and Tables

**Figure 1 cells-14-00447-f001:**
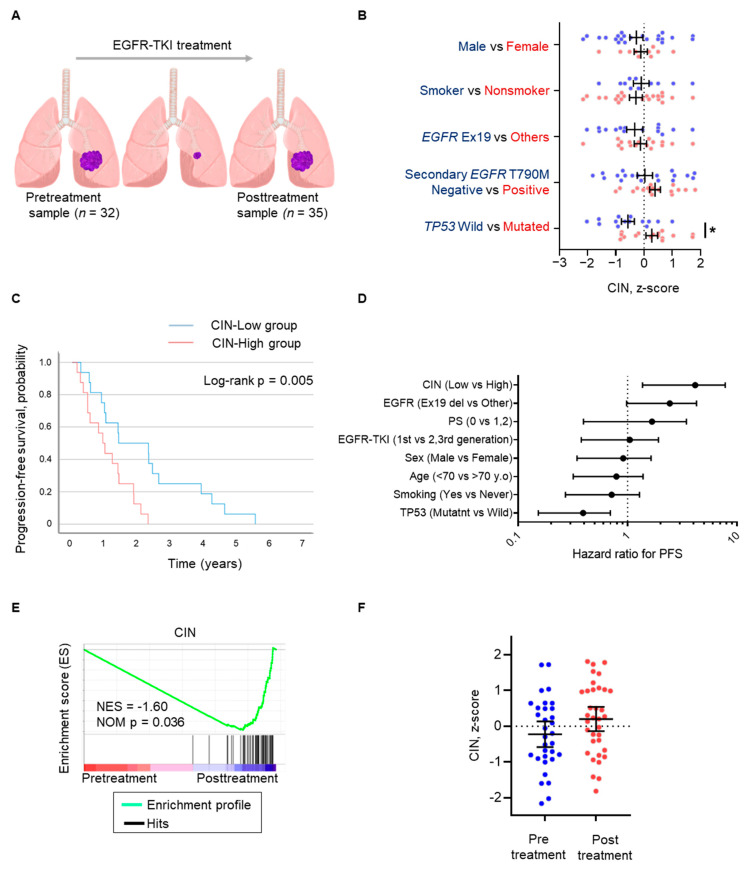
CIN in *EGFR*-mutated non-small-cell lung cancer is associated with the progression-free survival of EGFR-TKIs. (**A**) Schema of sampling at pretreatment (*n* = 32) as the “Pretreatment sample” and after the acquisition of EGFR-TKI resistance (*n* = 35) as the “Posttreatment sample”. (**B**) Scattered plot of CIN z-scores of individuals pre- and posttreatment samples. Clinical characteristics are analyzed as indicated. Data represent the mean ± SEM. (**C**) Progression-free survival curves for patients treated with EGFR-TKIs, separated into CIN-high and CIN-low groups. (**D**) Cox proportional hazards model for progression-free survival for EGFR-TKI adjusted by factors comprising CIN, type of *EGFR* mutation, performance status, EGFR-TKI generation, sex, age, smoking, and *TP53* mutation. (**E**) Gene set enrichment analysis (GSEA) plots showing the gene enrichment pattern of CIN between pretreatment and posttreatment samples. Normalized enrichment score (NES) and *p* value are shown on the plot. (**F**) Scattered plot of CIN z-scores of individual samples is shown and compared between pre- and posttreatment. Data represent the mean ± SEM. * *p* < 0.05; CIN, chromosomal instability; EGFR-TKI, epidermal growth factor receptor tyrosine kinase inhibitor; SEM, standard error of the mean.

**Figure 2 cells-14-00447-f002:**
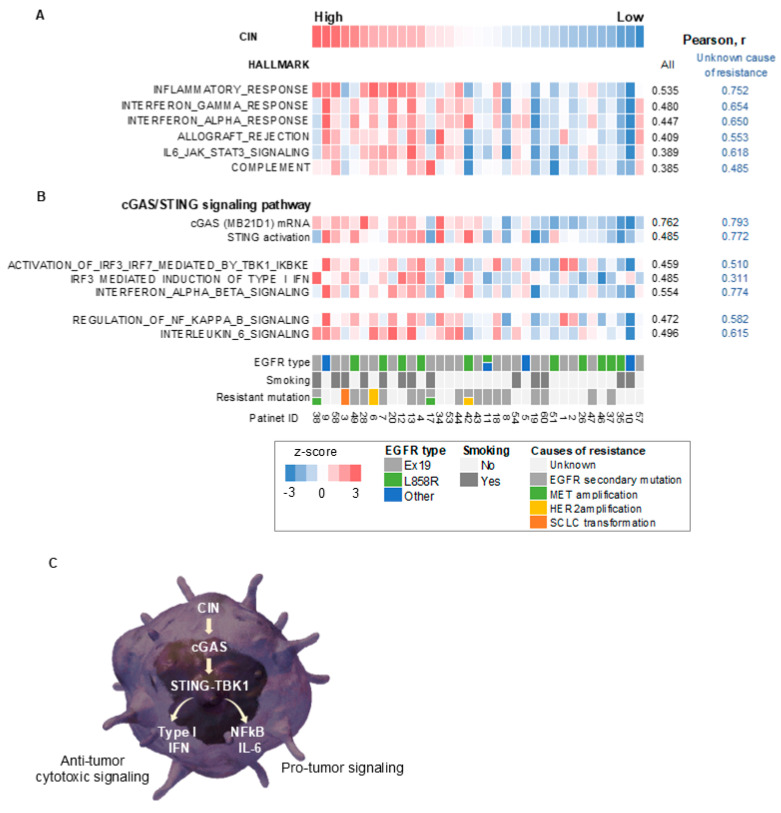
CIN correlated with cGAS–STING and its downstream signaling in *EGFR*-mutated non-small-cell lung cancer with acquired resistance to EGFR-TKIs. (**A**) A heatmap sorted by CIN z-score of hallmark-based gene signatures in the EGFR-mutated tumors with acquired resistance to EGFR-TKIs. (**B**) A heatmap sorted by CIN z-score of gene expression or gene signatures for cGAS–STING and its downstream signaling in the *EGFR*-mutated NSCLC tumors that have acquired resistance to EGFR-TKIs. (**C**) A schematic diagram showing a signaling pathway thought to be caused by CIN in *EGFR*-mutated NSCLC tumors with acquired resistance to EGFR-TKIs. CIN, chromosomal instability; EGFR, epidermal growth factor receptor; r, Pearson correlation coefficient; IL-6, interleukin 6; STING, stimulator of interferon response cGAMP interactor 1; TBK1, TANK-binding kinase 1; IFN, interferon; cGAS, cyclic GMP–AMP synthase; NF-κB, nuclear factor-kappa B.

**Figure 3 cells-14-00447-f003:**
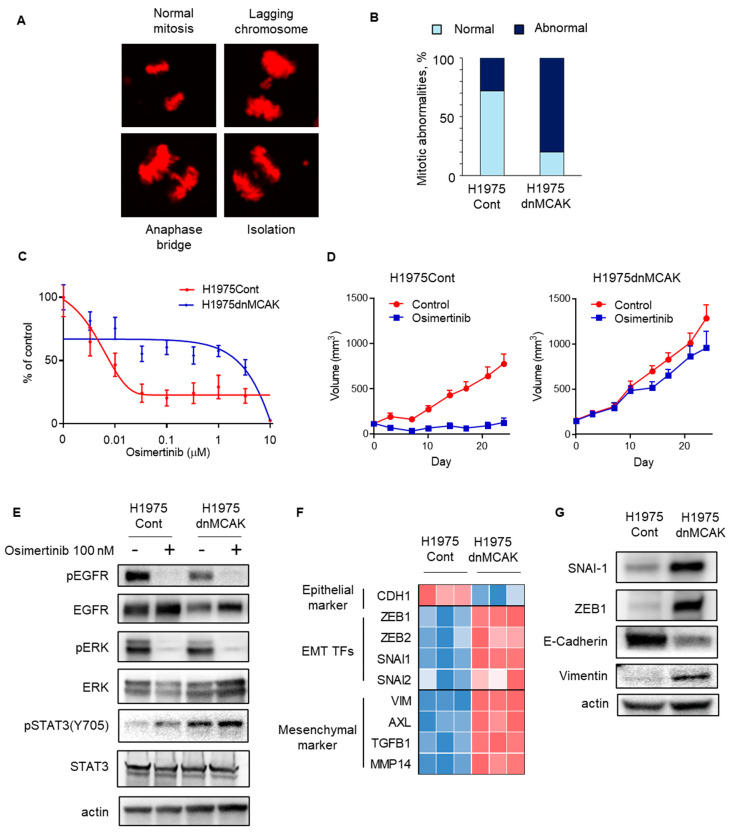
CIN-induced EGFR-TKI resistance in EGFR-mutated NSCLC may be associated with EMT. (**A**) A representative figure shows normal and abnormal chromosomes, including lagging chromosome, anaphase bridge, and isolation, in H1975dnMCAK cells. (**B**) A ratio of mitotic abnormalities in H1975Cont and H1975dnMCAK cells is shown. (**C**) In vitro growth inhibition assay showing CIN-induced EGFR-TKI resistance. H1975Cont or H1975dnMCAK cells were treated with the indicated concentrations of EGFR-TKI osimertinib for 3 d. Cell viability is shown relative to untreated control cells (mean ± SD of 6 replicate and 2 independent experiments). (**D**) In vivo study showing CIN-induced EGFR-TKI resistance. In a mouse xenograft model using H1975Cont or H1975dnMCAK cells, tumors were treated with osimertinib 5 mg/kg or vehicle 5 times per week. Tumor growth curves are shown. Each group consisted of eight mice. Data are the mean ± SEM. (**E**) Cellular signaling analysis for CIN-induced, EGFR-mutated NSCLC under EGFR-TKI exposure. H1975Cont or H1975dnMCAK cells were treated with 100 nM osimertinib for 3 h and then probed for the indicated proteins. (**F**) A heatmap of an epithelial marker, EMT-TFs, and mesenchymal markers in three clones of H1975Cont and H1975dnMCAK cells. (**G**) Immune blotting in H1975Cont or H1975dnMCAK cells of the indicated EMT-related proteins. EGFR, epidermal growth factor receptor; EMT-TFs, epithelial–mesenchymal transition transcription factors.

**Figure 4 cells-14-00447-f004:**
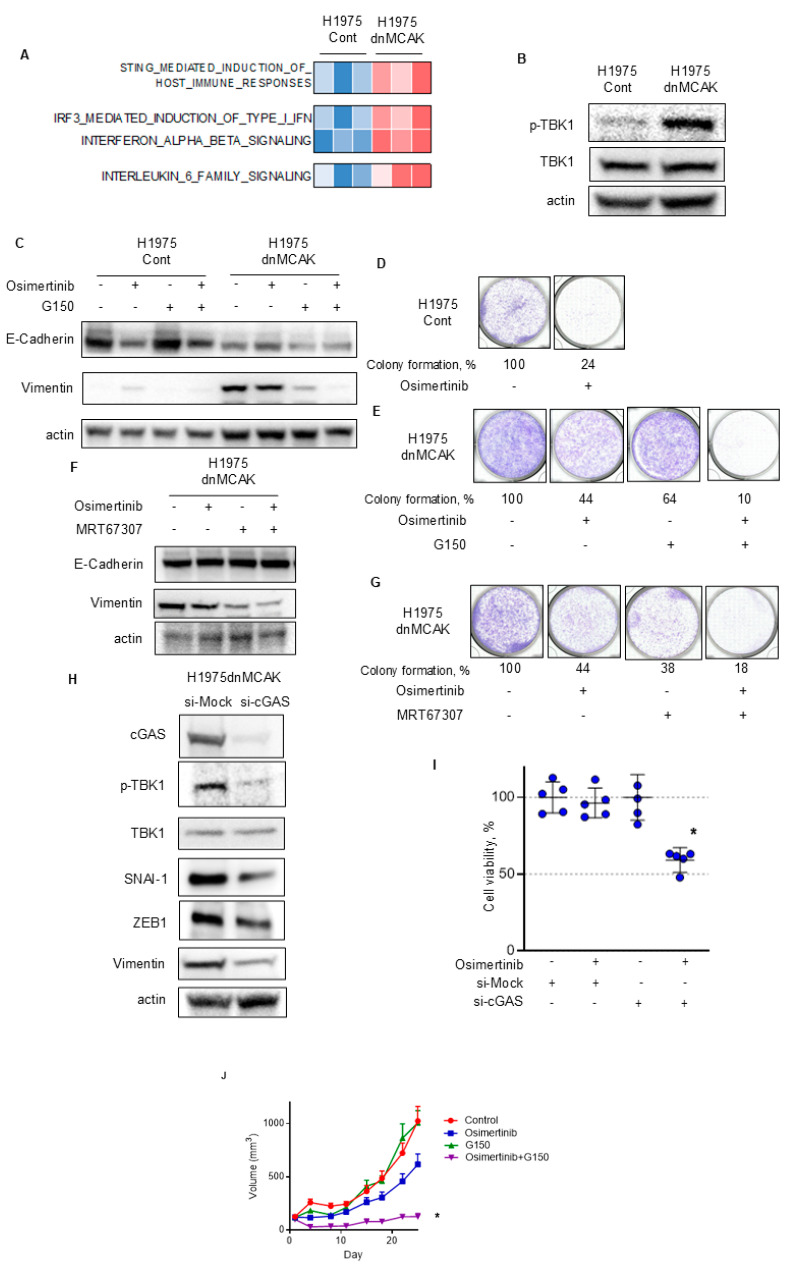
cGAS–STING–TBK1 signaling is associated with EMT in chromosomally unstable EGFR-activating NSCLC that is resistant to EGFR-TKIs. (**A**) A heatmap sorted by CIN z-score of Reactome pathway analysis, including STING, type I IFN, and IL-6 signaling. (**B**) CIN-dependent activation of TBK1 in EGFR-mutated NSCLC. Immune blotting of the indicated proteins in H1975Cont or H1975dnMCAK cells. (**C**) The cGAS inhibitor G150 reversed EMT associated with CIN. H1975Cont or H1975dnMCAK cells were treated with 100 nM osimertinib, 10 μM G150, or combination of these agents for 48 h and then probed for the indicated proteins. (**D**,**E**) The cGAS inhibitor G150 decreased EGFR-TKI resistance associated with CIN. Clonogenic assay for H1975Cont (**D**) and H1975dnMCAK (**E**) cells treated with/without 100 nM osimertinib, 10 μM G150, or a combination of both drugs for 10 d. The colony-formation area was measured and quantified using Image J. (**F**) The TBK1 inhibitor MRT67307 reversed EMT associated with CIN. H1975dnMCAK cells were treated with 100 nM osimertinib, 1 μM MRT67307, or a combination of these agents for 48 h and then probed for the indicated proteins. (**G**) The TBK1 inhibitor MRT67307 decreased EGFR-TKI resistance associated with CIN. Clonogenic assay for H1975dnMCAK cells treated with/without 100 nM osimertinib, 1 μM MRT67307, or a combination of both drugs for 10 d. (**H**) cGAS repression using siRNA reversed EMT associated with CIN. H1975dnMCAK cells were transiently transfected with siRNA targeting cGAS or a mock siRNA (si-Mock) and probed for the indicated proteins. (**I**) cGAS repression using siRNA decreased EGFR-TKI resistance associated with CIN. H1975dnMCAK cells were transiently transfected with siRNA targeting cGAS or mock siRNA (si-Mock) and treated with 100 nM osimertinib for 72 h. Cell viability is shown relative to untreated control cells (mean ± SD of five replicates). (**J**) The cGAS inhibitor G150 increase the susceptibility to EGFR-TKI osimertinib in chromosomally unstable EGFR-mutated NSCLC in vivo. In a mouse H1975dnMCAK xenograft model, tumors were treated with osimertinib 5 mg/kg or vehicle 5 times per week with/without G150 (5 mg/kg every 2 d). Tumor growth curves are shown. Each group consisted of seven mice. Data are the mean ± SEM. * *p* < 0.05; STING, stimulator of interferon response cGAMP interactor; cGAS, cyclic GMP–AMP synthase; TBK1, TANK-binding kinase 1.

**Figure 5 cells-14-00447-f005:**
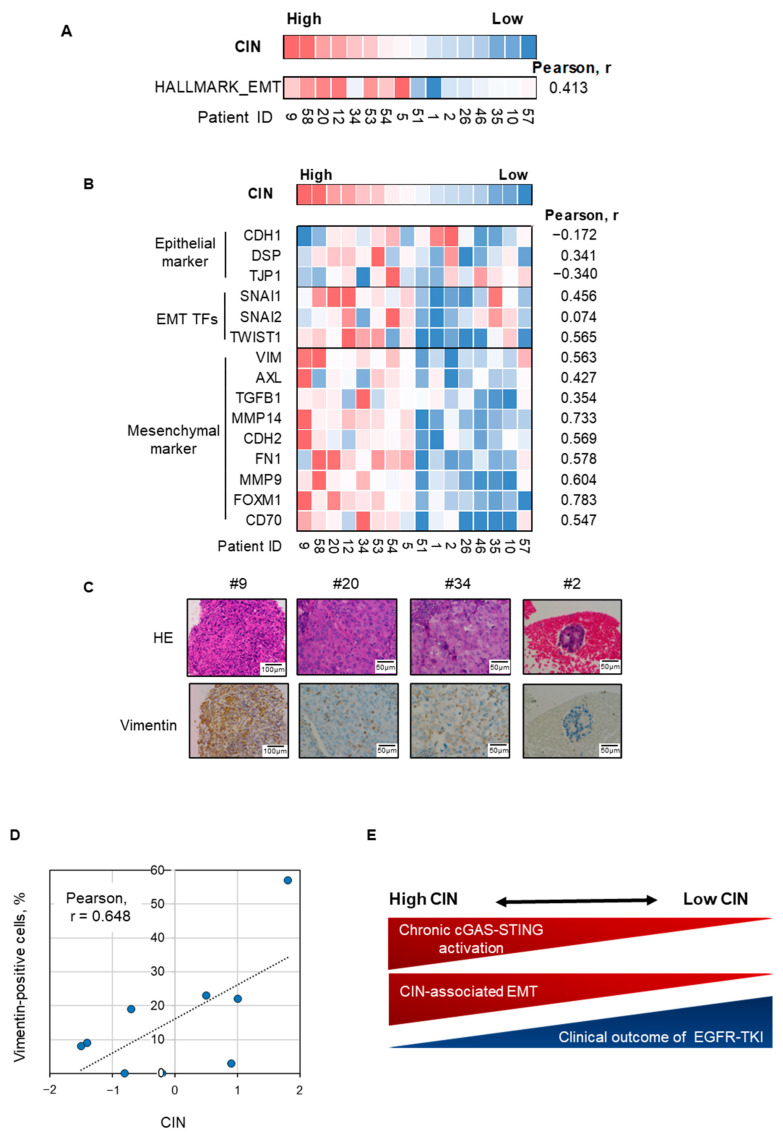
EMT is correlated with CIN in *EGFR*-mutated NSCLC tumors with acquired EGFR-TKI resistance. (**A**) A heatmap sorted by CIN z-score of Hallmark gene signature for epithelial–mesenchymal transition in *EGFR*-mutated NSCLC tumors with acquired EGFR-TKI resistance. (**B**) A heatmap sorted by CIN z-score of epithelial markers, EMT TFs, and mesenchymal markers as indicated. (**C**) Representative images of hematoxylin–eosin staining and vimentin immunohistochemistry for four *EGFR*-mutated NSCLC tumors with acquired resistance to EGFR-TKIs. (**D**) CIN is correlated with the ratio of vimentin-positive cells in *EGFR*-mutated NSCLC tumors. Correlation of CIN z-score and vimentin positive ratio in nine *EGFR*-mutated NSCLC tumors with acquired resistance to EGFR-TKIs. (**E**) Schematic figure of the relationship between CIN and cGAS–STING activation, inflammation, EMT, and sensitivity to EGFR-TKIs. CIN, chromosomal instability; EGFR-TKI, epidermal growth factor receptor tyrosine kinase inhibitor; NSCLC, non-small-cell lung cancer; EMT-TFs, epithelial–mesenchymal transition transcription factors.

## Data Availability

The data generated in this study are available upon request from the corresponding author.

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
