# Peer review of "Chromosomal Instability Is Associated with cGAS–STING Activation in EGFR-TKI Refractory Non-Small-Cell Lung Cancer"

_cells, 2025, doi:10.3390/cells14060447_

Round 1

Reviewer 1 Report

Comments and Suggestions for Authors

The MS by Yonesaka et al. regards the correlation between CIN and resistance to EGFR-TKIs in EGFR-mutated (EGFRm+) NSCLC. The Authors’ data (generated in vitro, using an EGFR-mutated NSCLC cell line, and in vivo utilizing a mouse xenograft model subcutaneously injected with this cell line) indicate that CIN activates the cGAS-STING pathway, which can mediate the TKI resistance via induction of EMT in NSCLC cells. The subject is potentially interesting and relevant for the readers of Cells as it sheds light on EGFR-TKI resistance, especially for cases without obvious on- or off-target genomic alterations that may explain it.

Yet, some revision of the work seems necessary to fully deserve publication.

SPECIFIC POINTS

Abstract

Line 30-35, “We demonstrated that CIN activates cGAS–STING signaling pathway, which leads to EGFR-TKI refractoriness in a subset of EGFR-mutated NSCLC. Specifically, CIN is associated with poor outcomes of EGFR-TKI treatment in patients with EGFR-mutated NSCLC. Transcriptome analysis revealed that CIN is associated with cGAS–STING signaling activation in EGFR-mutated NSCLC tumors.”: these 3 sentences are correct, but somehow redundant (essentially, they say the same thing). The authors could perhaps condense them in 1-2 sentences without losing the meaning.

Materials and Methods and/or Results

Paragraph 2.1 and/or 3.1: A more thorough description of the patients included in the study is needed. For instance, a table with demographic features of the pts should be considered. Especially important for the paper’s context is to specify the tumor stage of the pts and what EGFR-mutations the pts had. Also, which EGFR-TKIs they received (all the same drug(s) or different treatments among the pts?) should be specified. And in which line of treatment the EGFR-TKIs were used.

Furthermore, it is not clear whether the 35 samples obtained after acquisition of EGFR-TKI resistance are from the same 32 patients that provided the samples before EGFR-TKI treatment or from other pts. It should be specified to avoid misunderstandings. If it is the case, it should also be specified how many patients provided more than 1 sample after acquiring TKI resistance (32 vs 35). Finally, it should be indicated whether the samples after acquisition of TKI-resistance were re-biopsies from new metastases or from relapse/progression of the primary lung tumor.

Paragraph 2.2, line 113, “sufficient quantity of tumor cells (≥ 50% for RNA, ≥ 30% for DNA)”: In this respect, the minimum absolute number of required tumor cells should also be specified.

Paragraph 2.5: The H1975 cell line should briefly be described.

Paragraph 2.8, line 159-160: For clarity, it should be specified what the drugs are/do (for ex., that G150 is a selective human cGAS inhibitor; C176 a selective STING inhibitor and so on). It is only mentioned first in the Results.

Results

Fig. S1: The title of the figure is: “Pie graph about acquired resistant mechanisms for EGFR-TKI in EGFR-mutated non-small-cell lung cancer”. It would be more appropriate to slightly modify it, for instance as “Pie graph showing mechanisms of acquired resistance to EGFR-TKIs in EGFR-mutated non-small-cell lung cancer”.

Moreover, in the fig. S1, percentages for each resistance mechanism should be shown, and T790M alone, HER2 amp alone and HER2 amp + T790M should be shown separately to avoid confusion.

Line 270, TBK1 and IRF3: “TANK-binding kinase 1” and “interferon regulatory factor 3” should also be written in the text, not only in legend to fig. 2.

Moreover, the statement “and the activity of its downstream counterpart IRF3” is a bit confusing, it'd be better to specify that IRF3 is a substrate of/is phosphorylated by TBK1.

Line 299-301, “In a mouse xenograft model study, H1975Cont tumors shrunk upon osimertinib exposure, whereas H1975dnMCAK tumors did not shrink and continued to grow for three weeks (Fig. 3D).”: A representative picture of H1975cont tumors vs H1975dnMCAK tumors should be shown for visual comparison. It would help to understand better the results in the graphs in fig. 3D, i.e., the magnitude of tumor inhibition or lack hereof by Osimertinib in the mouse xenografts.

Line 304-306, “In contrast, STAT3 phosphorylation was greater in H1975dnMCAK cells than in H1975Cont cells and was maintained under EGFR-TKI exposure (Fig. 3E).”: there seems also to be more STAT3 protein in H1975dnMCAK cells than H1975cont cells, thus the claim could be explained by that. Some explanation/comment by the Authors is needed here.

The Authors often use the words “suppressed expression” and “suppressed colony formation” (for ex., line 357, 360, 364, 365): this is not completely correct, given that they are referred to things that, as it seems from the results, are not completely suppressed. “Decreased”, “reduced”, “downregulated” would be more appropriate terms.

Paragraph 3.5, line 417-419, “The highest expression of vimentin was observed in tumor #9, which had the highest CIN score among these tumors, and partial vimentin expression was observed in tumors with moderate CIN, including tumors #20 and #34 (Fig. 5C and 5D).”: high expression of Vimentin by IHC would also be expected in tumor #58, given that it too has highest CIN score and highest expression of vimentin RNA on the heat map. For the sake of for reproducibility and validation, it would be appropriate if the authors could show high vimentin expression by IHC in this tumor as well.

Discussion

Line 498-500, “Indeed, in H1975dnMCAK cells, there were fluctuations in the expression of multiple genes associated with EMT and cross-talk with NF-κB signaling (fig. S6).”: This should be shown more clearly on figure S6, as it is important to support the Authors' hypothesis that NF-κB, instead of IL-6 or JAK, is involved in the CIN-associated EMT in EGFRm+ NSCLC.

Line 501-502, “In the current study, CIN or its downstream cGAS or STING expression was correlated with the clinical outcome of EGFR-TKI treatment (Figure 1C, fig. S7a, b).”: It should be explained why Fig. 7s shows worse PFS for “STING low” than “STING high”, while “cGAS low” has better PFS than “cGAS high”. Is that due to STING inducing IFN-I pathway, which is not involved in causing EGFR-TKI resistance, as the Authors have hinted in other parts of their MS?

Line 516-8, “Moreover, the current study did not assess the influence of CIN-dependent cGAS-STING signaling activity in EGFR-mutated NSCLC on the tumor immune environment or the efficacy of EGFR-TKI therapy.”: The phrase might need to be reformulated more clearly: For instance, "Yet, the current study did no assess ..." could be more suitable. Additionally, the effect of cGAS-STING on efficacy of EGFR-TKI treatment has been in part investigated in the mouse xenograft model (Fig. 4J).

The results in vitro and in the mouse xenograft model have all been generated with one cell line, the H1975. Have the Authors tried to reproduce their results using other EGFRm+ NSCLC cell lines? It seems to be a major limitation of the study and should be at least discussed together with other study limitations.

References

References 2 and 41 have an incomplete title.

Comments on the Quality of English Language

Some sentences or terminology require modification, see Comments for and Suggestions for Authors for details.

Author Response

Dear reviewer 1

We appreciate your kind and valuable comments on our manuscript. We here provide detailed replies to your suggestions and comments.

Your comments: Line 30-35, “We demonstrated that CIN activates cGAS–STING signaling pathway, which leads to EGFR-TKI refractoriness in a subset of EGFR-mutated NSCLC. Specifically, CIN is associated with poor outcomes of EGFR-TKI treatment in patients with EGFR-mutated NSCLC. Transcriptome analysis revealed that CIN is associated with cGAS–STING signaling activation in EGFR-mutated NSCLC tumors.”: these 3 sentences are correct, but somehow redundant (essentially, they say the same thing). The authors could perhaps condense them in 1-2 sentences without losing the meaning.

Our response: These sentences have been revised as follows and added to line 31 of the paper.

“Our observations indicate that CIN is associated with poor outcomes of EGFR-TKI treatment in patients with EGFR-mutated NSCLC, and the activated cGAS–STING signaling pathway in these tumors.”

Your comments: Paragraph 2.1 and/or 3.1: A more thorough description of the patients included in the study is needed. For instance, a table with demographic features of the pts should be considered. Especially important for the paper’s context is to specify the tumor stage of the pts and what EGFR-mutations the pts had. Also, which EGFR-TKIs they received (all the same drug(s) or different treatments among the pts?) should be specified. And in which line of treatment the EGFR-TKIs were used.

Our response: We added a new Supplementary Table S1 which showed patient characteristics such as age, Performance Status, type of EGFR mutations, EGFR-TKI. All patients received EGFR-TKI as 1st line therapy for advanced disease. This information can be found on line 100-101 of the paper.

Your comments: Furthermore, it is not clear whether the 35 samples obtained after acquisition of EGFR-TKI resistance are from the same 32 patients that provided the samples before EGFR-TKI treatment or from other pts. It should be specified to avoid misunderstandings. If it is the case, it should also be specified how many patients provided more than 1 sample after acquiring TKI resistance (32 vs 35). Finally, it should be indicated whether the samples after acquisition of TKI-resistance were re-biopsies from new metastases or from relapse/progression of the primary lung tumor.

Our response: Of all samples, 23 cases were paired specimens before and after EGFR-TKI treatment. In each case, only one re-biopsy was performed after developing resistance to EGFR-TKI. These points are described in lines 97-99 of the paper. Furthermore, the site of the re-biopsy and whether it is a new lesion is also listed in the new Supplementary Table S2.

Your comments: Paragraph 2.2, line 113, “sufficient quantity of tumor cells (≥ 50% for RNA, ≥ 30% for DNA)”: In this respect, the minimum absolute number of required tumor cells should also be specified.

Our comments: This time, instead of the minimum absolute number of tumor cells, the criteria were set for the percentage of cancer cell content in the tissue and the amount of nucleic acid. For DNA analysis, the percentage of cancer cells should be more than 30% and the required amount of DNA should be 110 ng. For transcriptional profiling, the percentage of cancer cells should be more than 50% and the required amount of RNA should be 20 ng. These are described in lines 113 of the main text and lines 5-6 and 31-32 of the Supplementary Methods.

Your comments: Paragraph 2.5: The H1975 cell line should briefly be described.

Our comments: We described the following statement in line 137 of this paper as you suggested.

H1975 cells harboring EGFR L858R/T790M double mutation (RRID:CVCL_1511) were co-transfected with……..

Your comments: Paragraph 2.8, line 159-160: For clarity, it should be specified what the drugs are/do (for ex., that G150 is a selective human cGAS inhibitor; C176 a selective STING inhibitor and so on). It is only mentioned first in the Results.

Our comments: We described the following statement in line 159-161 of this paper as you suggested.

“After 24 h, cells were treated with 10 µM G150, a selective cGAS inhibitor, 0.5 µM C176, a selective STIG inhibitor, 1 µM MRT67307, an IKKε and TBK-1 inhibitor, 0.1 µM osimertinib, or a combination of these drugs.”

Your comments: Fig. S1: The title of the figure is: “Pie graph about acquired resistant mechanisms for EGFR-TKI in EGFR-mutated non-small-cell lung cancer”. It would be more appropriate to slightly modify it, for instance as “Pie graph showing mechanisms of acquired resistance to EGFR-TKIs in EGFR-mutated non-small-cell lung cancer”.

Our comments: As you indicated, the title of Fig. S1 has been changed as follows.

“Pie graph showing mechanisms of acquired resistance to EGFR-TKIs in EGFR-mutated non-small-cell lung cancer (n = 35).”

Your comments: Moreover, in the fig. S1, percentages for each resistance mechanism should be shown, and T790M alone, HER2 amp alone and HER2 amp + T790M should be shown separately to avoid confusion.

Our comments: As you suggested, I have included the percentage of each resistance mechanism in supplementary Fig. S1.

Your comments: Line 270, TBK1 and IRF3: “TANK-binding kinase 1” and “interferon regulatory factor 3” should also be written in the text, not only in legend to fig. 2.

Our comments: As you suggested, I have described “TANK-binding kinase 1” and “interferon regulatory factor 3” in line 271.

Your comments: Moreover, the statement “and the activity of its downstream counterpart IRF3” is a bit confusing, it'd be better to specify that IRF3 is a substrate of/is phosphorylated by TBK1.

Our comments: As you suggested, we described the following statement in line 270-272 of text.

“Furthermore, CIN correlated with the activity of TANK-binding kinase 1 (TBK1), a binding partner of STING, and its substrate, interferon regulatory factor 3 (IRF3), as well as with type I IFN signaling (Fig. 2B).”

Your comments: Line 299-301, “In a mouse xenograft model study, H1975Cont tumors shrunk upon osimertinib exposure, whereas H1975dnMCAK tumors did not shrink and continued to grow for three weeks (Fig. 3D).”: A representative picture of H1975cont tumors vs H1975dnMCAK tumors should be shown for visual comparison. It would help to understand better the results in the graphs in fig. 3D, i.e., the magnitude of tumor inhibition or lack hereof by Osimertinib in the mouse xenografts.

Our comments: Indeed, it might have been more impressive to show a picture of the tumor. However, in this case, we did not take pictures of the tumors because we judged that showing the change in tumor size over time would be convincing enough.

Your comments: Line 304-306, “In contrast, STAT3 phosphorylation was greater in H1975dnMCAK cells than in H1975Cont cells and was maintained under EGFR-TKI exposure (Fig. 3E).”: there seems also to be more STAT3 protein in H1975dnMCAK cells than H1975cont cells, thus the claim could be explained by that. Some explanation/comment by the Authors is needed here.

Our comments: Reviewing the blotting images, the STAT3 image was not clear. Therefore, we performed Western blotting again with the remaining lysate, which showed STAT3 more clearly. For this reason, we would like to replace Fig. 3E.

Your comments: Paragraph 3.5, line 417-419, “The highest expression of vimentin was observed in tumor #9, which had the highest CIN score among these tumors, and partial vimentin expression was observed in tumors with moderate CIN, including tumors #20 and #34 (Fig. 5C and 5D).”: high expression of Vimentin by IHC would also be expected in tumor #58, given that it too has highest CIN score and highest expression of vimentin RNA on the heat map. For the sake of for reproducibility and validation, it would be appropriate if the authors could show high vimentin expression by IHC in this tumor as well.

Our comments: Sample #58 had a small amount of tumor because it was collected by bronchoscopy. Also, sample #58 was insufficient to perform IHC because it was used in multiple studies and there were only a few remaining. Therefore, IHC could not be performed on sample #58.

Your comments: Line 498-500, “Indeed, in H1975dnMCAK cells, there were fluctuations in the expression of multiple genes associated with EMT and cross-talk with NF-κB signaling (fig. S6).”: This should be shown more clearly on figure S6, as it is important to support the Authors' hypothesis that NF-κB, instead of IL-6 or JAK, is involved in the CIN-associated EMT in EGFRm+ NSCLC.

Our comments: Based on your suggestion, I have added the following statement regarding TGFB1 in lines 501-504 as one hypothetical example.

“For example, Huber et al. reported that TGF-β–dependent induction of EMT depends at least in part on NF-κB activity. The TGFB1 expressed more in H1975dnMCAK cells than in H1975Cont cells and may be involved in EMT (fig. S6).”

Your comments: Line 501-502, “In the current study, CIN or its downstream cGAS or STING expression was correlated with the clinical outcome of EGFR-TKI treatment (Figure 1C, fig. S7a, b).”: It should be explained why Fig. 7s shows worse PFS for “STING low” than “STING high”, while “cGAS low” has better PFS than “cGAS high”. Is that due to STING inducing IFN-I pathway, which is not involved in causing EGFR-TKI resistance, as the Authors have hinted in other parts of their MS?

Our comments: Based on your suggestion, I have added the following statement regarding STING or cGAS expression-associated PFS in lines 506-513 and added Supplementary Figure S8.

“cGAS mRNA expression was correlated with CIN in EGFR-mutated non-small cell lung cancer (Figure 2B). Reportedly, cGAS is an IFN-stimulated gene, and its overexpression is associated with Type I IFN activation [50]. This suggests that cGAS expression may serve as a surrogate marker for CIN that potentially activates Type I IFN. Additionally, STING mRNA expression showed an inverse correlation with CIN, suggesting a potential negative regulation of STING mRNA expression by excessive CIN-induced STING activation (fig. S8). Thus, although hypothetical, STING mRNA expression may be a surrogated marker for CIN.”

Your comments: Line 516-8, “Moreover, the current study did not assess the influence of CIN-dependent cGAS-STING signaling activity in EGFR-mutated NSCLC on the tumor immune environment or the efficacy of EGFR-TKI therapy.”: The phrase might need to be reformulated more clearly: For instance, "Yet, the current study did no assess ..." could be more suitable. Additionally, the effect of cGAS-STING on efficacy of EGFR-TKI treatment has been in part investigated in the mouse xenograft model (Fig. 4J).

Our comments: Based on your suggestion, the text has been revised as follows in lines 527-529.

“Yet, the current study did not assess the influence of CIN-dependent cGAS-STING signaling activity in EGFR-mutated NSCLC on the tumor immune environment or the efficacy of EGFR-TKI therapy.”

Your comments: The results in vitro and in the mouse xenograft model have all been generated with one cell line, the H1975. Have the Authors tried to reproduce their results using other EGFRm+ NSCLC cell lines? It seems to be a major limitation of the study and should be at least discussed together with other study limitations.

Our comments: Thank you for suggestion. I have written this limitation on lines 530-532 as follows.

“Another limitation is that dnMCAK induction was successful in transducing CIN into the H1975 cell line, but not into other cell lines. For further studies, alternative methods of transducing CIN into multiple cell lines need to be tested.”

Your comments: References 2 and 41 have an incomplete title.

Our comments: Thank you for suggestion. I have revised the references.

Reviewer 2 Report

Comments and Suggestions for Authors

In this work, the authors investigate the role of CIN in driving resistance to EGFR-TKIs in EGFR-mutated NSCLC. They demonstrate that CIN activates the cGAS-STING signaling pathway, leading to resistance in some patients. Transcriptome analysis revealed a strong correlation between CIN and cGAS-STING activation, further promoting tumor progression through IL-6 signaling and EMT. Notably, blocking the cGAS-STING-TBK1 pathway reversed EMT and restored EGFR-TKI sensitivity, highlighting a potential therapeutic strategy to overcome resistance.

#The study highlights how CIN-driven cGAS-STING activation contributes to treatment failure in EGFR-mutated NSCLC.

#The findings suggest that targeting the cGAS-STING-TBK1 axis could restore drug sensitivity.

#This study strengthens the connection between CIN, immune response, and EMT-mediated drug resistance, offering new insights into tumor progression.

The manuscript is interesting, however, it can be improved and strengthened by addressing the following comments -

Minor points:

#The author should discuss potential inhibitors of the cGAS-STING pathway that could be clinically tested and their relevance in overcoming EGFR-TKI resistance in NSCLC.

#The authors may consider citing [PMID: 39125832], which explores the lncRNA/miRNA axis in EMT and drug resistance in NSCLC. This study provides a complementary perspective on EMT regulation and therapy resistance, which could strengthen the discussion.

#The author should briefly mention other known EGFR-TKI resistance mechanisms and how CIN fits into the broader landscape of resistance in NSCLC.

Author Response

Dear reviewer 2

We appreciate your kind and valuable comments on our manuscript. We here provide detailed replies to your suggestions and comments.

Your comments: #The author should discuss potential inhibitors of the cGAS-STING pathway that could be clinically tested and their relevance in overcoming EGFR-TKI resistance in NSCLC

Our response: Clinical development of cGAS or STING inhibitors is underway for inflammatory diseases, but has not yet been initiated for malignancies. This point is noted on line 533-535 below and reference 54 has been added.

“Despite some limitations, STING or cGAS inhibitors are currently in clinical development for inflammatory diseases and may be applicable to EGFR-mutated NSCLC [54].”

Your comments: # The authors may consider citing [PMID: 39125832], which explores the lncRNA/miRNA axis in EMT and drug resistance in NSCLC. This study provides a complementary perspective on EMT regulation and therapy resistance, which could strengthen the discussion.

Our response: Thank you for your kind suggestion. We described this point on lines 488-489 as below and added reference 46.

“Or SNAIL was reported to facilitate drug resistance and EMT [46].”

Your comments: #The author should briefly mention other known EGFR-TKI resistance mechanisms and how CIN fits into the broader landscape of resistance in NSCLC.

Our response: In the present study, we found that CIN and the associated cGAS-STING signaling activity were stronger in tumors with unknown resistance mechanisms than in tumors with previously reported resistance mechanisms after the acquisition of resistance to EGFR-TKIs. Thus, CIN may be more strongly involved in resistance independent of specific genetic abnormalities. This point is noted in lines 458-460.

Round 2

Reviewer 1 Report

Comments and Suggestions for Authors

The Authors have satisfactorily addressed most of the initial comments and revised their MS accordingly. However, there remain minor points to be clarified before full suitability for publication.

  • Abstract, line 31-33: The original three sentences have been revised by deleting too much text, thereby removing how the authors obtained the results highlighted in the sentence. It would be more understandable to write: "In this study, we demonstrate by transcriptomic analysis that CIN activates cGAS–STING signaling pathway, which leads to EGFR-TKI refractoriness in a subset of EGFR-mutated NSCLC patients".

  • Mat. & Meth:, paragraph 2.2: The Authors have replied to the previous comment by writing in their rebuttal that "instead of the minimum absolute number of tumor cells, the criteria were set for the percentage of cancer cell content in the tissue and the amount of nucleic acid. For DNA analysis, the percentage of cancer cells should be more than 30% and the required amount of DNA should be 110 ng. For transcriptional profiling, the percentage of cancer cells should be more than 50% and the required amount of RNA should be 20 ng. These are described in lines 113 of the main text and lines 5-6 and 31-32 of the Supplementary Methods". In this regard: A) for clarity, the required inputs/amounts of DNA and RNA should be specified in the main text for Mat. & Meth. (i.e., paragraph 2.2.), not as supplementary material; B) the minimum absolute number of tumor cells required in the biopsies for obtaining the indicated inputs of nucleic acids from these cells remain to be specified… As rule of thumb and depending on the sensitivity of methods, people use 200, 400, 500 etc. tumor cells as absolute number to make sure that enough DNA/RNA from tumor cells is obtained. Then obviously, the percentage of tumor cell content is also important, but there must be a minimum number of tumor cells for getting sufficient tumor DNA for the analyses. What is the absolute number of tumor cells used for minimum cut-off in the study?

  • Results: A representative picture of H1975cont tumors vs H1975dnMCAK tumors remains to be shown for proper visual comparison and confirmation of the data shown in graphic form in fig. 3D.

  • Discussion: line 500-501: What does "fluctuations in the expression of multiple genes" mean exactly? Fluctuations is a generic and dynamic term. Not clear how the Authors use it here. Do they mean that genes were first upregulated and then downregulated (to fluctuate = to move up and down or back and forth like a wave). Perhaps, the sentence should be rephrased more clearly from a scientific viewpoint.
  • Line 503: “The TGFB1 expressed more in …” should be “The TGFB1 was expressed in ..." 

Comments on the Quality of English Language

The authors have introduced some typos in the new text that they have added because of the revision. One ex. is “STIG inhibitor” on line 160, but there are others. The new text should therefore be checked for typos. The same for the new supplementary material (for ex. “Meidan” in Table S1).

Author Response

Dear reviewer 1

We appreciate your kind and valuable comments on our manuscript. We here provide detailed replies to your suggestions and comments.

Your comments: Abstract, line 31-33: The original three sentences have been revised by deleting too much text, thereby removing how the authors obtained the results highlighted in the sentence. It would be more understandable to write: "In this study, we demonstrate by transcriptomic analysis that CIN activates cGAS–STING signaling pathway, which leads to EGFR-TKI refractoriness in a subset of EGFR-mutated NSCLC patients".

Our response: The sentence has been revised as you suggested.

Your comments: Mat. & Meth:, paragraph 2.2: The Authors have replied to the previous comment by writing in their rebuttal that "instead of the minimum absolute number of tumor cells, the criteria were set for the percentage of cancer cell content in the tissue and the amount of nucleic acid. For DNA analysis, the percentage of cancer cells should be more than 30% and the required amount of DNA should be 110 ng. For transcriptional profiling, the percentage of cancer cells should be more than 50% and the required amount of RNA should be 20 ng. These are described in lines 113 of the main text and lines 5-6 and 31-32 of the Supplementary Methods". In this regard: A) for clarity, the required inputs/amounts of DNA and RNA should be specified in the main text for Mat. & Meth. (i.e., paragraph 2.2.), not as supplementary material; B) the minimum absolute number of tumor cells required in the biopsies for obtaining the indicated inputs of nucleic acids from these cells remain to be specified… As rule of thumb and depending on the sensitivity of methods, people use 200, 400, 500 etc. tumor cells as absolute number to make sure that enough DNA/RNA from tumor cells is obtained. Then obviously, the percentage of tumor cell content is also important, but there must be a minimum number of tumor cells for getting sufficient tumor DNA for the analyses. What is the absolute number of tumor cells used for minimum cut-off in the study?

Our response: As you suggested, we have moved the description about amount of nucleic acid from Supplementary Methods to the main text on line 117. Also, we clarified that the minimum absolute number of tumor cells required is 200, on line 114.

Your comments: Results: A representative picture of H1975cont tumors vs H1975dnMCAK tumors remains to be shown for proper visual comparison and confirmation of the data shown in graphic form in fig. 3D.

Our response: I understand very well that the points you mentioned are important and thank you for your advice. However, this time we did not take pictures of the tumor because we were able to graphically show a clear treatment effect.

Your comments: Discussion: line 500-501: What does "fluctuations in the expression of multiple genes" mean exactly? Fluctuations is a generic and dynamic term. Not clear how the Authors use it here. Do they mean that genes were first upregulated and then downregulated (to fluctuate = to move up and down or back and forth like a wave). Perhaps, the sentence should be rephrased more clearly from a scientific viewpoint.

Our comments: As you suggested, we have revised this sentence as follows and included it in line 502-504.

“Indeed, the expression of multiple genes associated with EMT and NF-κB signaling was greater in H1975dnMCAK cells than H1975Cont cells (fig. S6)”

Your comments: Line 503: “The TGFB1 expressed more in …” should be “The TGFB1 was expressed in ..."

Our comments: The sentence has been revised as you suggested.

Your comments: The authors have introduced some typos in the new text that they have added because of the revision. One ex. is “STIG inhibitor” on line 160, but there are others. The new text should therefore be checked for typos. The same for the new supplementary material (for ex. “Meidan” in Table S1).

Our comments: We have corrected the typo as you indicated. We have also checked other texts and found no additional typos.